# A False Trigger-Strengthened and Area-Saving Power-Rail Clamp Circuit with High ESD Performance

**DOI:** 10.3390/mi14061172

**Published:** 2023-05-31

**Authors:** Boyang Ma, Shupeng Chen, Shulong Wang, Lingli Qian, Zeen Han, Wei Huang, Xiaojun Fu, Hongxia Liu

**Affiliations:** 1Key Laboratory of Wide Band-Gap Semiconductor Materials and Devices of Education, The School of Microelectronics, Xidian University, Xi’an 710071, China; 13273711065@163.com (B.M.); hanzeen1212@163.com (Z.H.); huangweifirst@163.com (W.H.); hxliu@mail.xidian.edu.cn (H.L.); 2Chongqing Acoustic-Optic-Electronic Co., Ltd. of CETC & 24 Institute, Chongqing 401331, China; qlingli@163.com; 3National Key Laboratory of Integrated Circuits and Microsystems, Chongqing 401332, China; xjfu2000@163.com

**Keywords:** power clamp circuit, ESD, HBM, false trigger

## Abstract

A power clamp circuit, which has good immunity to false trigger under fast power-on conditions with a 20 ns rising edge, is proposed in this paper. The proposed circuit has a separate detection component and an on-time control component which enable it to distinguish between electrostatic discharge (ESD) events and fast power-on events. As opposed to other on-time control techniques, instead of large resistors or capacitors, which can cause a large occupation of the layout area, we use a capacitive voltage-biased p-channel MOSFET in the on-time control part of the proposed circuit. The capacitive voltage-biased p-channel MOSFET is in the saturation region after the ESD event is detected, which can serve as a large equivalent resistance (~10^6^ Ω) in the structure. The proposed power clamp circuit offers several advantages compared to the traditional circuit, such as having at least 70% area savings in the trigger circuit area (30% area savings in the whole circuit area), supporting a power supply ramp time as fast as 20 ns, dissipating the ESD energy more cleanly with little residual charge, and recovering faster from false triggers. The rail clamp circuit also offers robust performance in an industry-standard PVT (process, voltage, and temperature) space and has been verified by the simulation results. Showing good performance of human body model (HBM) endurance and high immunity to false trigger, the proposed power clamp circuit has great potential for application in ESD protection.

## 1. Introduction

With the development of integrated circuits (ICs), ESD protection has become the major concern regarding the reliability of IC products [1,2,3]. In order to solve this problem, researchers have proposed the gate-grounded NMOS (GGNMOS), the silicon-controlled rectifier (SCR) structures, and the RC-based power-rail clamp circuit, which can provide a low resistance path to achieving ESD protection without affecting the fragile core circuit [4,5,6,7,8,9]. Among them, the RC-based power-rail clamp circuit has become the mainstream protection method in full chip ESD protection due to its low trigger voltage and mature manufacturing process [4,5,6,7,8].

However, the RC-based power-rail clamp circuit is often falsely triggered by severe power noise or fast power-up events, which leads to the burning out of the clamp MOSFET, as shown in Figure 1. To improve immunity to false trigger and make sure it has sufficient turn-on time during ESD events, the hybrid triggering method combining static and transient efficiency has been proposed [9,10]. However, the sustained leakage current path in [9] causes massive unnecessary power consumption, while the circuit in [10] shows false trigger in fast power-on pulses with 20 ns rise time. Another improvement method uses the separation technique of a detection component and an on-time control component, which makes the clamping MOSFET turn-on time completely independent of the ESD-transient detection circuit [11,12,13]. Figure 2a shows the overall circuit structure design method to prevent false trigger, consisting of three parts: the detection component, the on-time control component, and the clamp device. The traditional power clamp circuit separating the detection and on-time control component proposed by Miller et al. is shown in Figure 2b [11], and the modified circuit with a current mirror proposed by Qi Liu et al. is shown in Figure 2c [12]. Both of the two ESD circuits here can achieve voltage clamping and prevent false trigger during fast power-on events. However, the large resistors and capacitors in these two ESD circuits will undoubtedly increase the layout, which can ultimately cause an increase in the manufacturing costs of ICs. Furthermore, there is still the possibility of false trigger in the case of high frequency and large amplitude noise disturbance.

Therefore, an area-saving power clamp circuit which has good immunity to false trigger is proposed in this paper. As opposed to other on-time control techniques, we use a capacitive voltage-biased p-channel MOSFET in the on-time control part to realize Mega Ohm-level large equivalent impedance. Simulation results verify that the proposed power clamp circuit is area-saving. Furthermore, it can achieve μs level transient turn-on time to fully release ESD stress, while the RC time constant of the detection part is only 10 ns to avoid most of the false trigger events. In addition, the circuit also has a low standby leakage current under normal power-on conditions.

## 2. The Proposed Power-Rail ESD Clamp Circuit

### 2.1. Structure of Proposed Circuit

In general, for common HBM ESD signals the rising edge is less than 10 ns [14,15]. Therefore, only a very small time constant (10 ns) is required for the usual ESD signal detection component. However, due to the requirements of the electrostatic discharge time, the RC time constant setting is usually relatively large, close to 1 μs. This causes false triggering during fast power-on events, resulting in an abnormal opening of the discharge circuit, which in turn leads to an increase in chip power consumption or even burnout. 

Figure 3 shows the proposed power-rail clamp circuit. Due to the separation of the detection component and the on-time control component, it has a relatively small RC time constant (10 ns) in the ESD detection component, which results in good immunity to false trigger and reduces leakage during power-on, as shown in the purple box in Figure 3a. The on-time control component consists of charging and discharging modules, as shown in the green box in Figure 3a. The charging module is composed of a small capacitor C2 (100 fF) and an nmos2, which is responsible for pulling the node B to a low level by charging the C2. The discharge module is composed of a p-channel MOSFET, C2, and nmos1, which is responsible for pulling the node B to a high level by discharging the C2. Particularly, the capacitance between the gate and drain of the nmos1 and the capacitance between the gate and source of the p-channel MOSFET form a capacitive voltage divider, which can provide gate voltage for the p-channel MOSFET. The equivalent schematic diagram of the capacitive voltage divider circuit is shown in Figure 3b. Combining Equations (1) and (2), the equivalent resistance (*r*_ds_) of MOSFET is inversely proportional to the difference between the gate voltage and the source voltage (*V*_gs_).
(1)$$I_{ds}=\mu C_{ox}\frac{W}{L}{\left(V_{gs}-V_{th}\right)}^{2}(1+\lambda V_{ds})$$
(2)$$r_{ds}=\varphi V_{ds}/\varphi I_{ds}$$

If the p-channel MOSFET is biased at a relatively high gate voltage, the p-channel MOSFET will be equivalent to a huge resistance (~10^6^ Ω) after ESD events are detected in the circuit. In this way, the discharge time of C2 can be prolonged, and the voltage of node B can be raised slowly to ensure that the ESD clamping MOSFET has enough turn-on time to discharge static electricity. Moreover, the leakage current of the on-time control component is negligible (nA level) because the nmos1 and nmos2 are always in the off state after normal power-on.

### 2.2. Principle of Operation

Figure 4a shows the voltage of the key node under a 1.8 V/100 μs power supply. When the normal power-on happens, the transient detection component will not be triggered due to the fact that there is no rapidly rising ESD signal. So, the voltage of node A is kept at a low level and the nmos2 is kept off all the time. Meanwhile, the p-channel MOSFET is turned on quickly with the gate-biased voltage provided by the capacitive voltage divider. So, the voltage of node B is pulled up to a high level through the p-channel MOSFET and C2. Then the voltage of node D is kept at a level of zero and the clamping MOSFET (the BIGMOS in Figure 3a) is kept off. That is to say, the proposed power clamp circuit can accurately identify the power signal and maintain a standby state while the internal circuit is working normally. 

Figure 4b shows the voltage of the key node under ESD events. When the ESD events happen, the transient detection component with a Rl· C1 time constant is triggered by an ESD signal, and the voltage of node A is pulled up to a high level to turn on the nmos2. At this moment, the p-channel MOSFET is turned on quickly with the gate-biased voltage provided by the capacitive voltage divider. Then the voltage of node B is pulled down to a low level and the voltage of node D is pulled up to a high level. So, the clamping MOSFET is turned on. 

A little time later (proportional to Rl· C1 time constant), the voltage of node A changes to a low level, and the nmos2 is turned off. Then the C2 is discharged through the p-channel MOSFET. While the p-channel MOSFET in the saturation region is equivalent to a large resistance, because its long channel is narrow and pinched-off it takes a relatively long time for node B to be pulled up to a high level. Finally, the voltage of node D is pulled down to a low level, which turns off the clamping MOSFET. Therefore, the proposed clamp circuit can quickly recognize the ESD signal and ensure sufficient time (μs level) to discharge static electricity.

## 3. Simulation and Results Discussion 

Comprehensive simulation tests were conducted to illustrate the advantage of the proposed structure. All the tests were carried out on a Cadence simulation test platform based on the 180 nm process.

### 3.1. The Circuit-Level TLP Test

The transient transmission line pulsing test (TLP) is specifically designed to be one of the most effective methods used to verify the protection level of ESD circuits. Here, a square wave with a rising time of 10 ns and a voltage amplitude of 0–5 V was used to simulate the TLP stress [16,17]. 

Table 1 shows the main parameters used in the traditional circuit, the modified circuit, and the circuit proposed above. To ensure that the ESD detection capabilities of the three structures are the same, we kept R1 and C1, which are 10 kΩ and 1 pf, respectively, consistent in the three circuits. R2 and C2 are the key parameters to control the electrostatic discharge time, depending on the circuit structure used. The width of the clamping MOSFET (*W*_mos_) was set to 2000 μm, which can provide the same low resistance path to achieve ESD protection. After applying the same TLP stress to the above three circuits, the test results are shown in Figure 5a. We can clearly see that the discharge times of both traditional and modified circuits are 480 ns and 710 ns, respectively. The proposed circuit increases discharge time to 870 ns, which is far longer than the previous structures. This means that the proposed power clamp circuit can discharge static electricity more fully than the other two structures.

In general, microsecond electrostatic discharge times are sufficient, but worse conditions, such as surge voltage and current, can occur in different application environments. The proposed voltage clamp circuit can ensure the electrostatic discharge at the microsecond level and realize the adjustable discharge time. Figure 5b shows the gate voltage of the clamping MOSFET varying from the W/L of the p-channel MOSFET during an ESD event. It can be seen that the turn-on time of the clamping MOSFET increases with the increase of channel length. This is because the longer the channel, the larger the equivalent resistance and the longer the discharge time. More importantly, the discharge time can be adjusted by controlling the gate-biased voltage of the p-channel MOSFET, which achieves equivalent resistance adjustability by changing the opening degree of the channel.

### 3.2. Area-Efficiency Evaluation

Figure 6 depicts layout views of the modified clamp circuits with a current mirror and the proposed circuit mentioned above in which the MOSFET has a default width of 2000 um. The area of the modified clamp circuits with a current mirror is 60 µm × 61 µm in Figure 6a, while the area of the proposed circuit is only 45 µm × 61 µm in Figure 6b. Due to the large equivalent resistance of the voltage-biased p-channel MOSFET in the proposed circuit, it greatly reduces the need for the capacitor C2 while replacing the huge R2 area. Therefore, compared to the modified circuit with a current mirror, at least 30% of the layout area is saved by the proposed circuit.

### 3.3. Circuit-Level ESD Test

The circuit-level ESD test was performed with the HBM to verify the effectiveness of the clamp circuit under ESD conditions. For a 4 KV HBM waveform, the peak current is 2.67 A ± 10% with a rise time of less than 10 ns and a duration of 120–180 ns [18]. The simulation circuit and current waveform of the HBM are shown in Figure 7. In Figure 7a, the *C*_esd_ and *R*_esd_ are the equivalent capacitance and equivalent resistance of the human body, respectively. Considering the parasitic effect, we have added capacitance *C*_p_ and inductance *L*_p_. The general range of *L*_p_ values is 5–12 μH and the range of *C*_p_ values is 1–4 pF. The simulated 4 kV HBM waveform is shown in Figure 7b, and it can be seen that the current rise time (*t*_r_) is less than 10 ns, with a peak value (*I*_p_) of approximately 2.67 A, which meets the HBM testing standard.

Figure 8 shows the response of VDD under the 4 KV HBM ESD test, and the impact of temperature variations on the proposed power clamp is considered. With the increase of temperature, the maximum clamping voltage of the proposed circuit increases gradually, and the clamping ability of the circuit decreases slightly (the clamping voltage increased from 6.2 V to 6.9 V). This is mainly because the increase in temperature leads to an increase in the equivalent resistance of the clamping MOSFET. On the whole, even under high or low temperature conditions (−40 °C to 125 °C), the proposed circuit can clamp the power supply voltage below 7 V during 4 KV HBM events and can quickly drop to below 4 V within 50 ns. This shows that the proposed circuit has a superior electrostatic clamping ability.

### 3.4. Immunity to the Fast Power Events

Figure 9 shows the gate voltage of the clamping MOS when fast power-on events happen. The fast power-up events of 1 μs/1.8 V, 200 ns/1.8 V, and 20 ns/1.8 V are simulated, respectively. As we can see in Figure 9, the faster the power supply is powered on, the higher the transient gate voltage of the clamping MOSFET, but the shorter the duration time of the gate voltage. Even for the worst case of a 20 ns/1.8 V fast power-on, the gate voltage of the clamping MOSFET is about 292 mV, far below the threshold voltage (*V*_th_). Therefore, the proposed power clamp circuit can effectively avoid false triggering during fast power-on events. This is mainly due to the small time constant (10 ns) of the detection component; the detection circuit can distinguish well between ESD events and fast power-on events.

### 3.5. Immunity to Noise Characteristics

High switching rates usually cause power supply noise, which can cause energy consumption and even falsely trigger the power clamp circuit. So, it is necessary to verify the immunity to power supply noise by simulating a pseudorandom pulse. The added noise has a frequency of 500 MHz and an amplitude of 0.6 V, which is considered the worst case. The noise characteristic of the proposed circuit is shown in Figure 10. It can be seen that the maximum leakage current of the clamping MOSFET is only about 1.2 mA and the duration time is very short. Therefore, the proposed clamp circuit can significantly mitigate high frequency and large amplitude noise disturbance.

### 3.6. The Low Leakage Characteristic

Figure 11 shows the leakage current of the clamping MOSFET under different temperature conditions. During a fast power-on event when the VDD rises from 0 to 1.8 V (1 μs/1.8 V), the peak leakage current of the clamping MOSFET does not exceed 1.9 μA. After the power-on, the leakage current quickly decreases to the nA level. Although the leakage current slightly increases with the increase of temperature, it is still within an acceptable range (below 60 nA at 125 °C). The results verify that the proposed power clamp circuit is low power, and the energy consumption of the circuit can be ignored after power-on.

## 4. Performance Comparisons

Table 2 shows the results of comparing the proposed circuit with the other two-stage separated circuits (separate detection component and on-time control component) mentioned above. In the case of ensuring the same ESD detection capability (the same R1 and C1) and electrostatic discharge pathway (the same width of clamping MOSFET (*W*_mos_)), the proposed circuit occupies the smallest layout area, only 2745 μm^2^. At the same time, the proposed circuit has extremely low static leakage and remains at around 31 nA after power-on.

Performance comparisons of the proposed clamp with the representative prior approaches are presented in Table 3. Regarding the trigger circuit (TC) area-reduction ratio in Table 3, the baseline circuit for comparison is the traditional transient circuit with an RC time constant of 100 ns. By using a capacitor-biased p-channel MOSFET to achieve equivalent large resistance, the proposed circuit obtains a higher area efficiency than most prior types. The transient response time of the proposed circuit at the μs level is sufficient for ESD current discharge. Additionally, the circuit demonstrated high false trigger immunity in the worst case of fast power-up pulses. The most important quality of the proposed clamp is the adjustable transient response time which suits various ESD protection scenarios, which is not achieved in existing hybrid triggering clamps. By comparison, the proposed circuit performs better than prior circuits and provides an excellent solution for on-chip ESD protection.

## 5. Conclusions

A power clamp circuit which has good immunity to false trigger was proposed in this paper. On the one hand, by utilizing the principle of capacitive voltage division, the voltage-biased p-channel MOSFET in the discharge module is equivalent to a huge resistance (~10^6^ Ω) after ESD events are detected in the circuit. Thus, microsecond discharge times can be easily achieved while avoiding the use of large resistors and capacitors. Compared with traditional circuits, the proposed circuit area savings is at least 30% (trigger circuit area savings is at least 70%). On the other hand, the proposed circuit has a strong ability to prevent false triggering, supporting a power-on time of as fast as 20 ns and withstanding high-frequency noises of 500 MHZ/0.6 V. In addition, when the internal circuit is working normally, the proposed circuit can maintain a standby state, and the low standby leakage current is only 31 nA, avoiding energy consumption. In ESD events, the clamping MOSFET can be turned on quickly, forming a low resistance path to fully discharge static electricity. Even under high or low temperature conditions (−40 °C to 125 °C), the proposed circuit can clamp the power supply voltage below 7 V during 4 KV HBM events and can quickly drop to below 4 V within 50 ns. Therefore, the proposed circuit exhibits good HBM endurance and high immunity to false trigger, which have great application potential in ESD protection.

## Figures and Tables

**Figure 1 micromachines-14-01172-f001:**
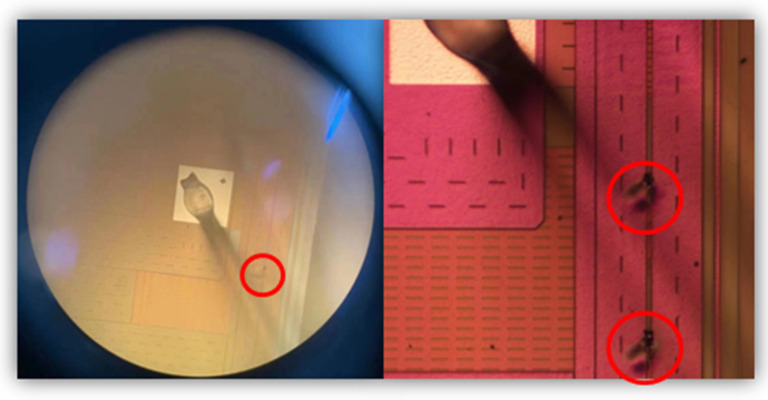
A picture of the burning out of the clamp MOSFET after testing. (The burnt places are circled in red).

**Figure 2 micromachines-14-01172-f002:**
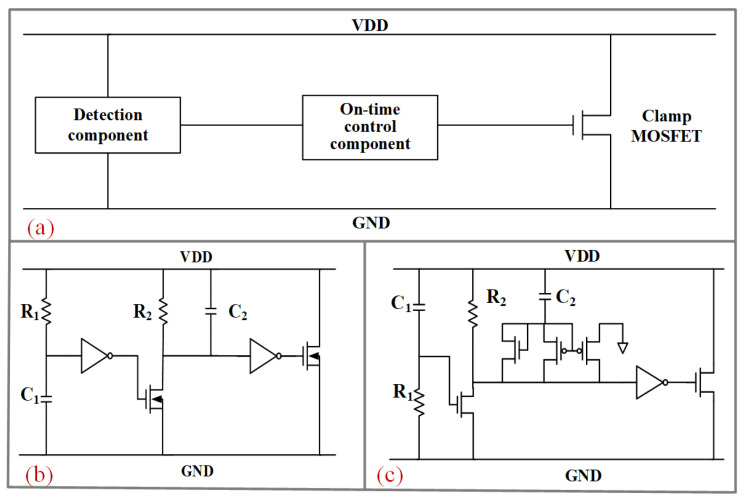
(**a**) Overall structure of a power supply clamp circuit. (**b**) The traditional power clamp circuit with separate detection and on-time control components [11]. (**c**) The modified circuit with current mirror [12].

**Figure 3 micromachines-14-01172-f003:**
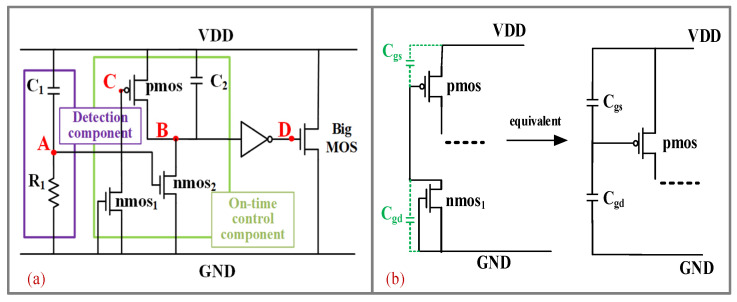
(**a**) The proposed power-rail clamp circuit. (**b**) The equivalent schematic diagram of the capacitive voltage divider circuit.

**Figure 4 micromachines-14-01172-f004:**
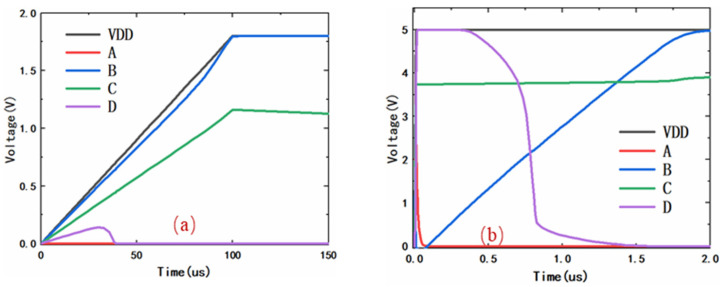
(**a**) The voltage of key node under 1.8 V/100 μs power supply. (**b**) The voltage of key node under ESD events (Simulated by 5 V/10 ns pulse waveform).

**Figure 5 micromachines-14-01172-f005:**
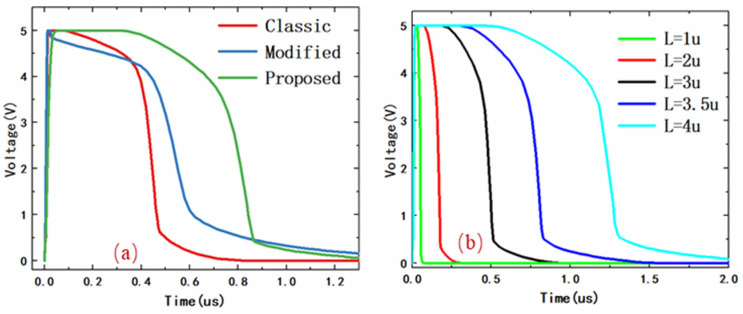
(**a**) The gate voltage of MOSFET of three circuits under circuit-level TLP test. (**b**) The gate voltage of MOSFET varying from the W/L of the p-channel MOSFET during an ESD event.

**Figure 6 micromachines-14-01172-f006:**
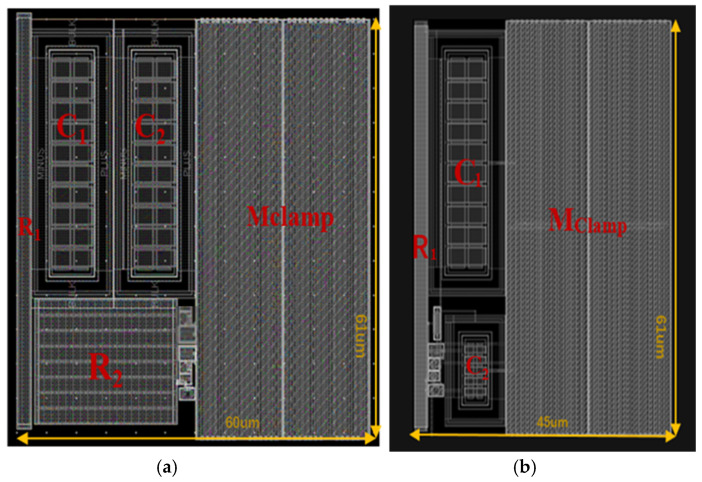
(**a**) Layout of the modified circuit with current mirror. (**b**) Layout of the proposed circuit.

**Figure 7 micromachines-14-01172-f007:**
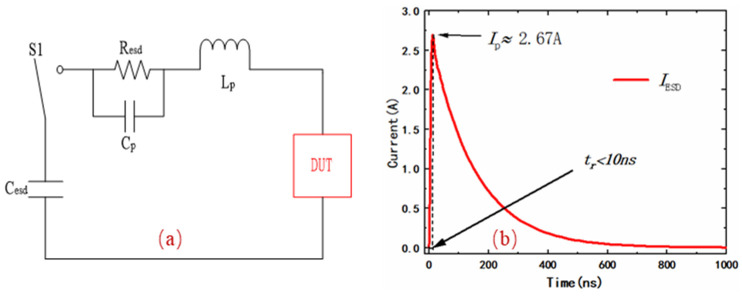
(**a**) The HBM for circuit-level ESD test. (**b**) Simulated HBM 4 kV discharge waveform.

**Figure 8 micromachines-14-01172-f008:**
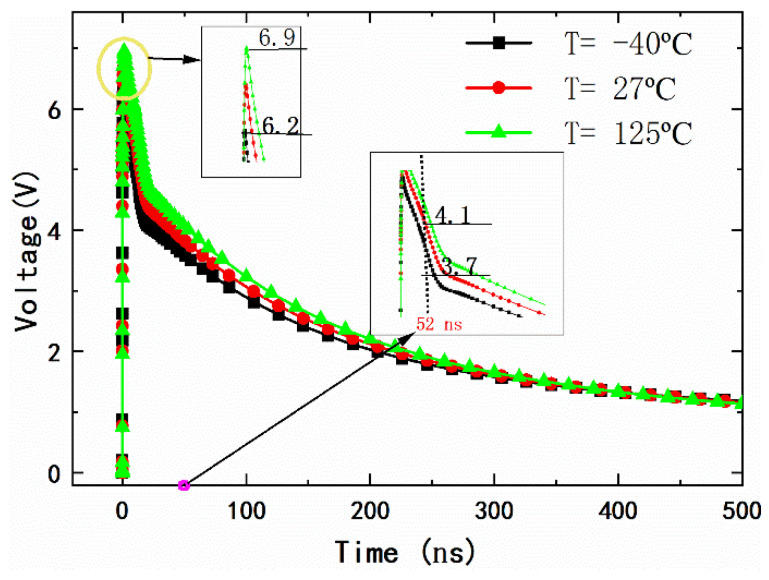
The response of VDD under circuit-level ESD test with 4 KV HBM.

**Figure 9 micromachines-14-01172-f009:**
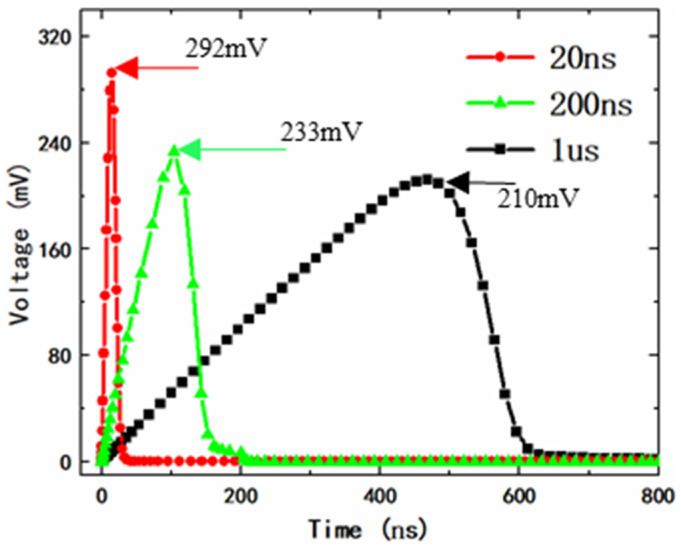
The gate voltage of MOSFET when fast-power-on events happen.

**Figure 10 micromachines-14-01172-f010:**
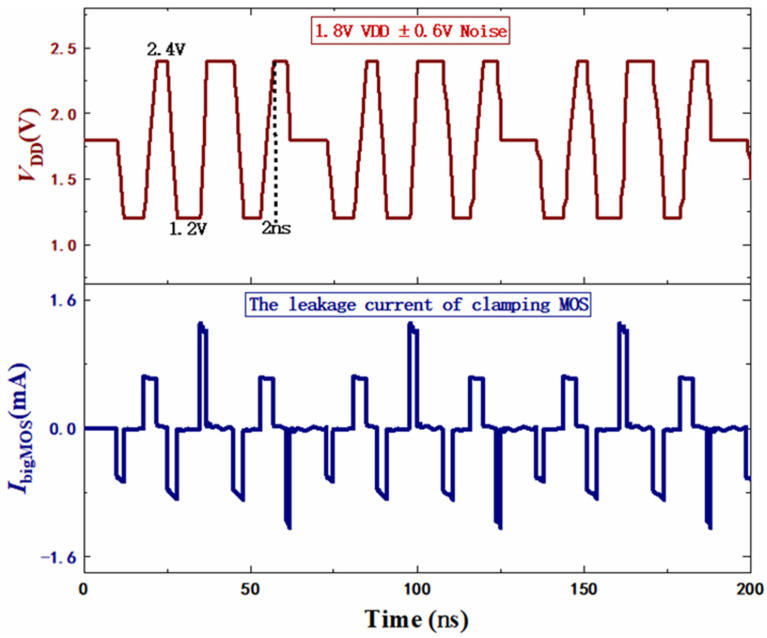
The respond of MOSFET under the disturbance of noise.

**Figure 11 micromachines-14-01172-f011:**
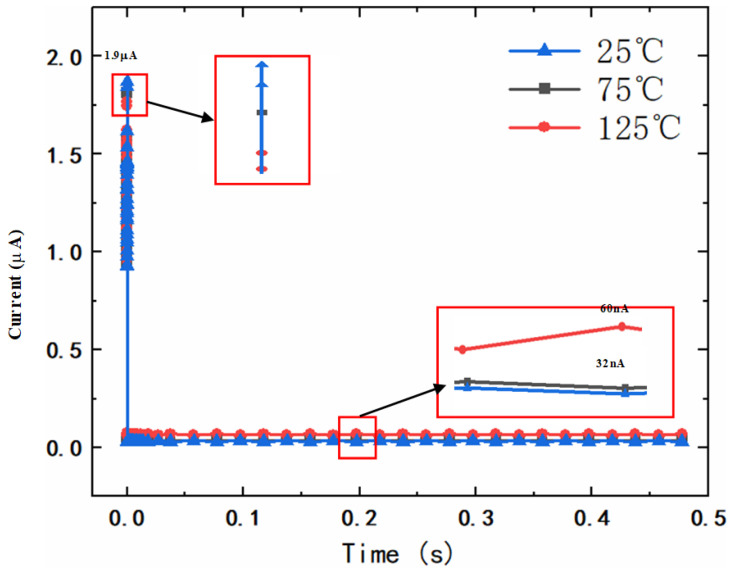
The leakage current under different operating conditions.

**Table 1 micromachines-14-01172-t001:** The main parameters of the three circuits mentioned above.

	The Classic [11]	The Modified [12]	The Proposed
R1	10 K	10 K	10 K
C1	1 p	1 p	1 p
R2	400 K	30 K	Voltage-biased MOSFET
C2	1 p	1 p	100 f
W_mos_	2000 μ	2000 μ	2000 μ

**Table 2 micromachines-14-01172-t002:** The comparison results of the three circuits mentioned above.

	The Classic [11]	The Modified [12]	The Proposed
Process	180 nm	180 nm	180 nm
W_mos_	2000 μ	2000 μ	2000 μ
Layout area	>5000	3660	2745
*I*_leak_ at 27 °C	31 nA	μA level	31 nA
False trigger	immune	immune	immune
HBM level	4 KV	4 KV	4 KV

**Table 3 micromachines-14-01172-t003:** Performance comparisons of the proposed clamp with the representative prior clamps.

	TED 2018 [16]	TDMR 2020 [19]	ISCAS 2022 [20]	TED 2022 [21]	The Proposed
Process	180 nm	BCD Process	28 nm	28 nm	180 nm
*I* _leak_	N/A	31 nA	7 nA	6.8 nA	31 nA
TC area-reduction ratio	No Reduction	No Reduction	~50% over the baseline circuit	~90% over the baseline circuit	>70% over the baseline circuit
False trigger	immune	immune	immune	immune	immune
Transient Response Time	100 ns	μs—level	μs—level	μs—level	μs—level
Adjustable time	NO	NO	YES	NO	YES

## Data Availability

Data are contained within the article.

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
