# Peer review of "A False Trigger-Strengthened and Area-Saving Power-Rail Clamp Circuit with High ESD Performance"

_micromachines, 2023, doi:10.3390/mi14061172_

Round 1
Reviewer 1 Report
Please refer to the attached file.

The text can be improved, with minor changes. Please revise it carefully.
Author Response
Dear reviewer, thank you for your suggestions and affirmations on this work. According to the comments, the manuscript has been revised one by one. The details are in the attachment.

Reviewer 2 Report
Dear Sir/Madam
My comments
1. Figure 4 needs some clarification to compare the results
2. The Authors did not tell us whether the discharge time of the proposed circuit in Figure 5 is good or not
3. Is there any relation between Lp and Cp in Figure 7a
4. How are the parameters identified in Table 1?
5. The Authors did not clarify whether there is an effect of temperature. Through Figure 8 (there is no temperature effect), is this possible??
6. Was the bigMOS response tested under noise perturbation at different frequencies or at a single frequency?
BR
Author Response

(The authors gave the same response as above.)

Reviewer 3 Report
The authors have presented a power clamp circuit with immunity to false trigger under fast power-on conditions in this manuscript. There are some modifications which should be considered in the manuscript as follows:
- It should be explained that how the device dimensions and also the circuit elements of the are calculated. It is suggested to provide proper analyses for the proposed circuit.
- The Conclusion part needs more explanation and details. Kindly explain the achievements of the proposed circuit in the conclusion part.
- There are several typos which should be corrected in the manuscript. For example: Abbreviations should be expanded at the first seen, such as ESD and HBM in the abstract. “PMOS” or “pMOS” are usually used for p-channel MOSFET not “pmos”. Check this sentence “The simulation circuit and current waveform of HBM are shown in the Figure 6.” In line 177. Figure 6 (a) and (b) in Figure 6 should be written as text under the figure, not embedded in the figure.
- Quality of figures should be improved, such that the text in the figures can be read easily.
- There are some works with the smaller layout area. It is suggested to add some extra works in the comparison table.
- The design procedure of the proposed clamp circuit is not clear. Kindly explain more about the design steps to describe that how the proposed circuit is obtained.
There are several typos which should be corrected in the manuscript. For example: Abbreviations should be expanded at the first seen, such as ESD and HBM in the abstract. “PMOS” or “pMOS” are usually used for p-channel MOSFET not “pmos”. Check this sentence “The simulation circuit and current waveform of HBM are shown in the Figure 6.” In line 177. Figure 6 (a) and (b) in Figure 6 should be written as text under the figure, not embedded in the figure.
Author Response

(The authors gave the same response as above.)

Round 2
Reviewer 3 Report
The authors have addressed all comments in the manuscript and the paper can now be accepted.